

# Protein dynamics insights from $^{15}N$-$^1H$ (TROSY) HSQC

Erik R.P. Zuiderweg

*University of Michigan Medical School, Department of Biological Chemistry, Ann Arbor, MI 41109, USA*

*Current address: 6585XW Mook, The Netherlands.*

*Correspondence to: Erik R.P. Zuiderweg (zuiderwe@umich.edu)*



**Abstract.** Protein dynamic information is customarily extracted from [15]N NMR spin-relaxation experiments. These experiments can only be applied to (small) proteins that can be dissolved to high concentrations. However, most proteins of interest to the biochemical and biomedical community are large and relatively insoluble. These proteins often have functional conformational changes, and it is particularly regretful that these processes cannot be supplemented by dynamical information from NMR.

We ask here whether (some) dynamic information can be obtained form the [1]H line widths in [15]N-[1]H HSQC spectra. Such spectra are widely available, also for larger proteins. We developed a computer program to predict amide proton line widths from (crystal) structures. As a calibration, we test our approach on BPTI. We find that we can predict most of the distribution of experimental amide proton line widths if we take the dipole-dipole interaction with at least 40 surrounding protons into account. When focusing our attention the outliers of the distribution, we find for BPTI a cluster

of conformationally broadened [1]HN resonances of residues in strands 10-15 and 36-40 of the beta sheet. Conformational exchange broadening of the [15]NH resonances for these residues was previously reported using [15]N relaxation measurements (Szyperski et al., J. Biomol. NMR 3, 151-164, 1993). There is little or no evidence for motional narrowing of the [1]HN resonances, also in agreement with earlier data using [15]N relaxation methods (Beeser et.al, J. Mol. Biol. 269, 154-164, 1997). We also apply our program to 42 kDa domain of the human Hsc70 protein. In this case, there is no

previous [15]N relaxation data to compare with, but we find, again from the outliers of the distribution, both exchange broadening and motional narrowing that appears to corroborate previous conformational insights for this domain.



## 1. Introduction


Providing evidence that proteins are dynamical rather than rigid molecules is a major contribution of solution NMR to structural biology and molecular biophysics (Kay, 1998). The dynamic information is extracted from NMR spin-relaxation experiments, mostly of the amide nitrogen (Kay et al., 1989), but also from methyls (Nicholson et al.,

1992) (Lee et al., 2000) and from carbonyl (Wang et al., 2006). The amide nitrogen relaxation experiments are the easiest to implement, require just $^{15}$N isotopic labeling, are potentially complete, and analysis software is broadly available (Mandel et al., 1995). But these experiments are rather insensitive (especially the $^{1}$H$\rightarrow$ $^{15}$N NOE) and can therefore only be applied to (small) proteins that can be dissolved to high concentrations.

However, most proteins of interest to the biochemical and biomedical community are large, and cannot be studied with

$^{15}$N dynamics measurements. This is particularly regretful, because larger proteins, more than smaller, often display functional conformational changes which cannot be supplemented by dynamical information from NMR. Moreover, considering the insight that fast dynamics contributes to configurational entropy (Akke et al., 1993) (Yang and Kay, 1996) (Lee et al., 2000), lack of measurement of dynamics also results in the lack of (experimental) understanding of the protein's thermodynamics. Due to the paucity of dynamical measurements of the proteins of interest to the biochemical

and biomedical community, the above "dynamics awareness" has not generally taken hold in that important area of science and medicine, which I find regretful.

However, $^{15}$N-$^{1}$H HSQC and TROSY HSQC experiments can be recorded for (very) large proteins ( < 300 kDa) at relatively low concentrations (< 50 uM) . This experiment contains conformational dynamics information in the intensity and line widths of its cross peaks. In this contribution, we explore if we can harvest (some) of the dynamical information

from that data, without the need for specific "relaxation" experiments and/or labeling strategies (Gardner et al., 1997).

The study of a protein by solution NMR usually starts by recording such a $^{1}$HN-$^{15}$N HSQC or TROSY HSQC spectrum of the sample. Such a dataset is seen as a "fingerprint" of the protein, from which important molecular parameters can immediately be gleaned. For instance, an experienced NMR spectroscopist recognizes from such a spectrum whether the sample is pure, the protein is (mostly) folded, whether it aggregates, or whether it has multiple

conformations. At the outset, the NMR spectroscopist expects that all cross peaks in a HSQC should be of about equal intensity. After all, the $^{1}J_{HN}$ scalar couplings vary very little, and transfer efficiencies in the (R)INEPT sequences are expected to be almost equal. If the cross peaks differ in greatly intensity, something else could be at play. Amide proton mass exchange, conformational exchange broadening or protein molecular dynamics are suspected.

Furthermore, the practicing NMR spectroscopist spends many weeks studying the details of the assignment and NOESY

spectra and is aware of many details in the data that may indicate dynamical effects. NMR resonances maybe narrower or broader than the average width, may be doubling, or expected cross peaks may be absent. Is there a way to understand and "harvest" this semi-quantitative information before it is lost when the (student) scientist moves on to the next project or job?




Let us take a look at the $^{15}N$-$^{1}H$ HSQC spectrum of bovine trypsin inhibitor (BPTI). BPTI spectra and assignments are available at the Biological Magnetic Resonance Bank, while a 1.3 Å resolution crystal structure (9PTI.PDB) is available in the Protein Data Bank. BPTI is considered to be a rigid protein, in order to resist proteolysis by trypsin itself. In Figure 1 we show a 500 MHz $^{15}N$-$^{1}H$ HSQC spectrum of BPTI, as downloaded as time-domain data from the Biological

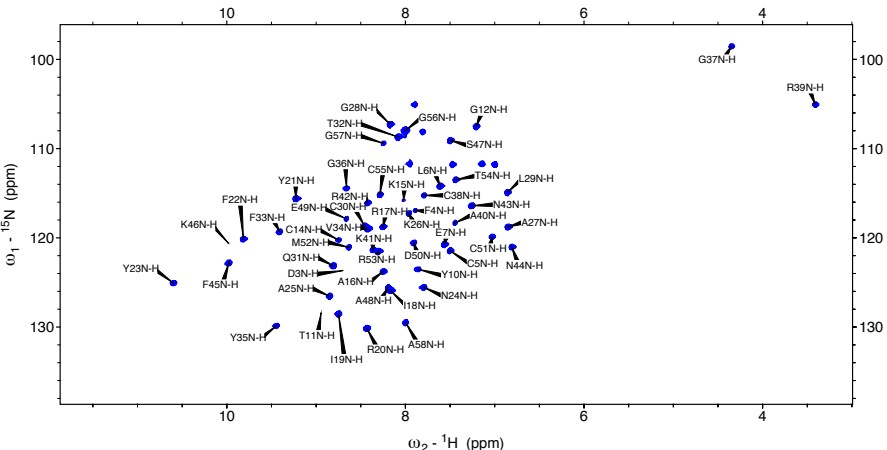

**Figure 1.** *500 MHz $^{15}N$-$^{1}H$ HSQC spectrum of Bovine BPTI at $30^{o}$ C, pH 5.8. (BMRB Entry 5307) Processed with 1 Hz EM in $t_2$, cosine in $t_1$ .*

Magnetic Resonance Data bank. We processed the data using nmrPipe (Delaglio et al., 1995) with a 1 Hz exponential window in $t_2$, in order to be as quantitative as possible for the following studies. At the contour level of Figure 1, several peaks are missing, indicating intensity dispersion. In Figure 2 we show the signal-to-noise ratio (S/N) of the cross peaks in this spectrum. We see that, even for BPTI, we have intensity differences of over a factor of 30. What are these differences due to?


The obvious answer is that the intensity differences arise from differences in the intrinsic $^{1}HN$ and $^{15}N$ line widths ($\delta\upsilon_{1/2}$) in the spectrum, even though that is not apparent from the contour plots. A modest spread in line widths causes a larger spread in peak intensity (S/N) as the latter should be proportional to:

$$S/N \sim \left\{ \exp\left(-\pi\delta\upsilon_{1/2}^{HN} / \left(2 \times {}^{1}J_{HN}\right)\right) \right\}^{M} \times \exp\left(-\langle t_1 \rangle \delta\upsilon_{1/2}^{N}\right) \times \frac{1}{\upsilon_{1/2}^{N}} \times \frac{1}{\upsilon_{1/2}^{HN}}$$
[1]

where $\langle t_1 \rangle$ is (some) average of the $t_1$ acquisition time. The term in the curly brackets stems from signal loss during the (R)INEPT transfers in the sequence. *M* corresponds to the number of (R)INEPT periods in the HSQC sequence; here 3,





because a sensitivity-enhanced HSQC sequence was used. The $^1$HN line widths effective in the (R)INEPT transfer

periods include (for this dataset) the (unresolved) $^3J_{HNH\alpha}$ couplings, but not magnetic field inhomogeneity. Indeed, when

using equation [1] we obtain a reasonable correlation between the tabulated S/N ratio and the S/N calculated from the

raw $^1$HN line widths in the spectrum (see Figure 3).

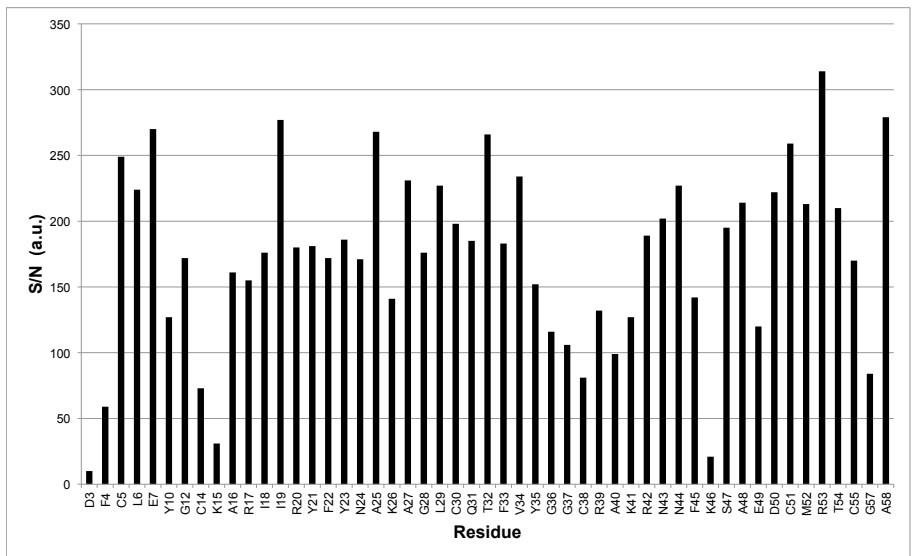

**Figure 2.** *Signal-to-noise ratios (S/N) of the cross peaks in Figure 1*

What causes the variation in $^1$HN line width underlying this intensity dispersion? There are many possibilities.

First , the (unresolved) $^3J_{HN-H\alpha}$ scalar coupling present in this data varies between 0 and 10 Hz. Second, the dipolar

proton environment of each amide proton varies. Third, anisotropic rotational diffusion may cause differences in the $^1$H

line widths. Fourth, the amide proton lines could be life-time broadened by mass exchange with water. Fitfh, last but not

least, local dynamics could affect the line widths, either by narrowing (fast local dynamics) or broadening

(conformational exchange dynamics on the ms – us timescale), e.g. by aromatic ring flipping, di-sulfide isomerization, or

general conformational flexibility.

We will address these points one by one. Can anisotropic rotational diffusion cause the line width differences? BPTI is

an ellipsoid with 8 Å and 16 Å for the short and long axes respectively. We calculate from the classical Woessner

equations (Woessner, 1962) that $R_2$ varies +/- 12 % when considering different angles from a relaxation vector to the

diffusion axes. But that is for individual relaxation vectors – the $^1$HN-$^1$HX relaxation vectors contributing to the dipolar

$R_2$ relaxation of a particular $^1$HN point in different directions; so, in practice, the small orientational effects will mostly

cancel.

The intrinsic (unprotected) amide proton exchange rate is given by the empirical relation (Englander et al., 1972) :



$$k_{ex} = \frac{\ln 2}{200}\left[10^{pH-3} + 10^{3-pH}\right] \times 10^{0.05T} \qquad [2]$$

where $T$ is in $^0$C and $k_{ex}$ in min$^{-1}$.

From the experimental parameters of the spectrum (30 $^o$C, pH 5.8) we calculate a 1.15 s$^{-1}$ exchange rate, giving rise to a broadening of ~ 0.3 Hz for unprotected amide proton resonances. Amide protons engaged in H-bonds within the protein will exchange much slower, with even less broadening. We find that variation in amide proton exchange is not significant for this spectrum.

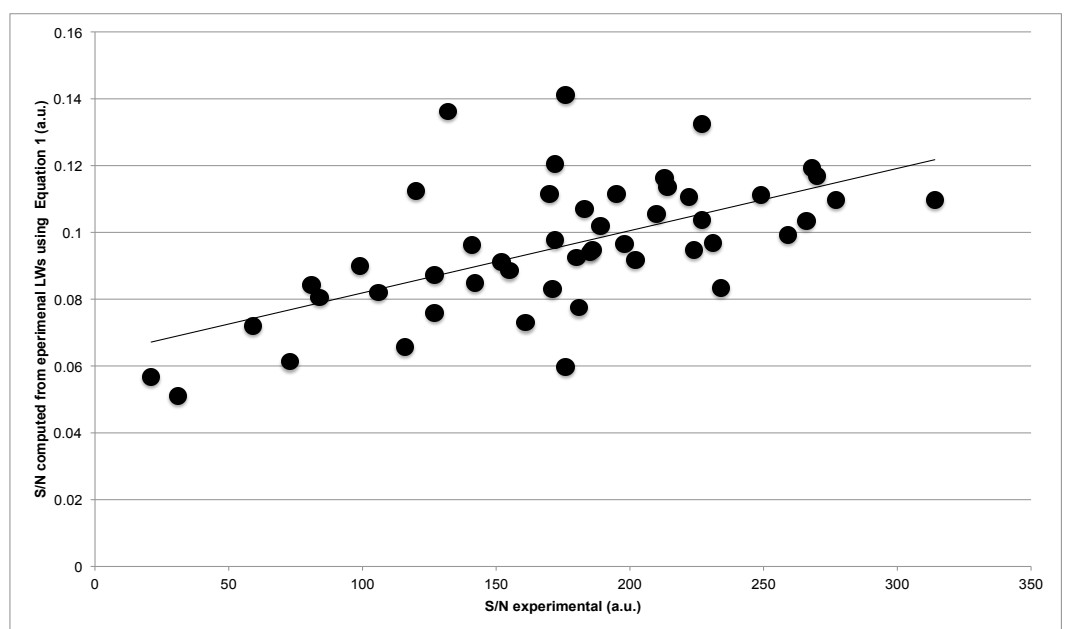

**Figure 3.** *S/N computed from the experimental $^1$HN line widths, according to Eq 1. $R^2$=0.36.*

In principle, the specific proton environment of each amide proton is known from the (high resolution) structure of BPTI. Hence, the dipolar R$_2$ relaxation of $^1$HN due to its surrounding protons can be calculated, given the three-dimensional structure. The unresolved $^3J_{HN-H\alpha}$ scalar coupling is also knowable and can be obtained from the structure as well, using the following Karplus equation (Lee et al., 2015)

$$^3J_{HN-H\alpha} = 8.83\cos^2(\theta - 60) - 1.29\cos(\theta - 60) + 0.20 \qquad [3]$$

where $\theta$ is the dihedral angle spanned by C'-N-Ca-C'.

Summarizing, we have a handle on variables 1 through 4 that affect the $^1$HN line width. Hence, making a calculation of these variables, and comparing the resulting calculated line width with the experimental (reduced, see below) line width, should uncover the presence (or not) of the dynamic properties of the protein, in a sequence-specific fashion.





## 2. Results and Discussion

The $R_2$ relaxation rate for like spins is given by (Goldman, 1988)

$$R_2^{H(H)} = \frac{1}{20}\left(\frac{\mu_0}{4\pi}\frac{\gamma_H\gamma_H\hbar}{r_{HH}^3}\right)^2 \times \left\{9\tau_c + \frac{15\tau_c}{1+\omega_H^2\tau_c^2} + \frac{6\tau_c}{1+\left(2\omega_H\right)^2\tau_c^2}\right\} \qquad [4]$$

where $\mu_0$ is the permittivity of space, $\gamma$ are the gyromagnetic ratios, $\hbar$ Planck's constant divided by $2\pi$, $\omega$ the resonance frequency, and $\tau_c$ the rotational correlation time. The reported rotational correlation time for BPTI is according to NMR, 3.25 ns (Beeser et al., 1997), 3.5 ns (Sareth et al., 2000), and 2.5-3.5 ns by MRD studies (Gottschalk et al., 2003), while the predicted value from an empirical equation (Daragan and Mayo, 1997) is 3.5 ns. We chose an average of 3.4 ns.

BPTI contains many protons that interact magnetically. In our programs, described in the appendix, we find typically that 40 protons are present in a 6 Å sphere around an amide proton. All $R_2$ relaxation rates of these $N$ other protons $j$ for an amide proton $i$ will co-add if the relaxation vectors $ij$ diffuse independently from each other:

$$R_2^{i-total} = \sum_{j\neq i}^{j=N} R_2^{ij} \qquad [5]$$

Obviously the assumption underlying equation [5] cannot be correct, because the interacting protons in a protein are *not* diffusing independently. One has to consider dipole-dipole cross-correlated $R_2$ relaxation (also called interference). However, we can show that relaxation interference is mostly canceled in multi-spin systems, and can be neglected as a source for large deviations of Eq. [5] (see Appendix).

We thus move on by using equation 4 and 5 to calculate the dipolar line widths of [1]HN from the coordinates of BPTI and compare the results with the experimental line widths. Before we can make that comparison, we have to correct the experimental [1]HN line widths for a few more effects. We need to consider dipolar relaxation due to the amide nitrogen, amide proton chemical shift anisotropy and field inhomogeneity. We estimate that the [15]N-[1]HN dipolar interaction accounts for 3 Hz, that the [1]H CSA contributes 1 Hz at 500 MHz, while field inhomogeneity typically is limited to 1 Hz. What the exact values are may be disputed, but they are small and approximately constant for all amides, or partially cancel in the TROSY version of the HSQC. The current experiment is a classical HSQC, for which we estimate 5 Hz combined for all amide protons. We thus subtract 5 Hz from the apparent experimental [1]HN line widths as obtained from careful two-dimensional peak-fitting using Sparky (Goddard and Kneller, 2000).

We compute the scalar couplings from the crystal structure using the Karplus equation [3]. The obtained values were subtracted from the experimental line widths. What is left is what we call the "reduced experimental line width", which *should* consist of just the sum of the [1]HN-[1]HX dipolar line widths, potentially affected by the fast and/or slow dynamics we try to uncover. Of course, we could go the other way around, rather than subtracting above factors from the experimental data, adding all these effects to the calculated line width. However, just the variation in [3]J$_{HNHA}$ will then dominate that sum and make the comparison between experiment an calculation rather flattered and meaningless.



For the computations we use the crystal structure of BPTI (9PTI), which was refined to a resolution of 1.3 Å. The proton coordinates were added by the routine Molprobity (Williams et al., 2018). For Figure 4 we used Equations [4] and [5], taking into account all protons in a sphere of 6 Å around the individual amide protons. Such a sphere typically contains 40 protons. Taking a larger sphere into account does not significantly change the results (see Table 1).


| Sphere (Å) | Sum of [1]HN line widths (Hz) |
|---|---|
| 3 | 269.7 |
| 4 | 311.1 |
| 5 | 329.8 |
| 6 | 337.1 |
| 7 | 340.3 |
| 8 | 342.0 |
| 9 | 342.9 |
| 10 | 343.4 |

**Table 1.** *Sum of calculated dipolar [1]HN linewidths (Eqs. 4 and 5) for BPTI as a function of the radius of the sphere around the amide protons.*





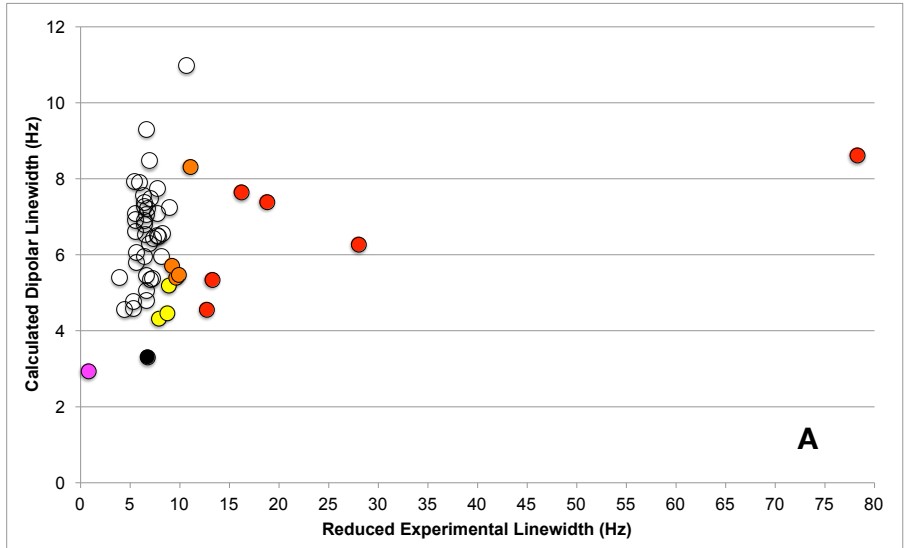

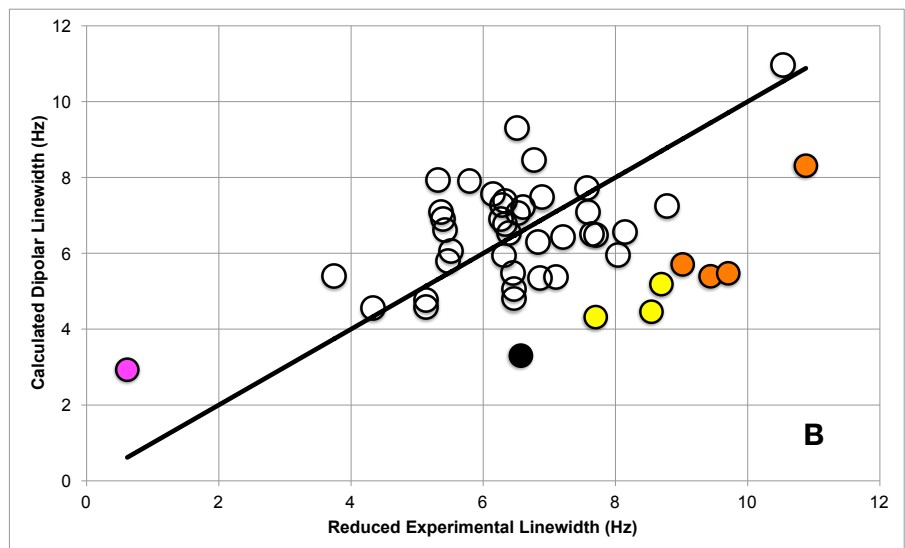

**Figure 4.** *Calculated dipolar $^1HN$ linewidths (Eq 4 and 5) vs. the Reduced Experimental $^1HN$ line width for BPTI. Panel B is an enlargement of panel A. The line in panel B is y=x.*

*For both panels: the red points correspond to residues 3,11,14,15,37 and 46 ; the orange points to residues 4,16,36 and 40 ; the yellow points to residues 10,12 and 14; the black point is residue 44; the magenta point is residue 58 (C-terminus).*

Regretfully, as one can see in Figure 4A, there is basically *no correlation* between the reduced experimental and calculated values. But given as it is, we may still learn something from the comparison. We see that calculated widths fall in the range 3-11 Hz, but that the reduced experimental widths cover a range from 2 – 80 Hz (Asp 3HN is 80 Hz). We suggest that all experimental resonances with line widths over 11 Hz must be broadened by conformational dynamics, (colored red, orange and yellow in Figure 4a and Figure 4b) and under 3 Hz by local fast motion (colored magenta).



In Figure 5 we show the putatively broadened and narrowed [1]HN resonances on the structure of BPTI. The magenta point in Figures 4A and 4B corresponds to the C-terminal residue. In [15]N relaxation studies (Beeser et al., 1997) , this residue has a strongly reduced order parameter. The red points of Figure 4A and 4B are also plotted on the structure in Figure 5A, where we find most of them in the "lower left side" of BPTI. This area comprises two anti-parallel beta strands with residues 10-15 and 36-40, and harbors a Cys14 – Cys38 disulfide. In Figure 4b we show an enlargement of

the central area of Figure 4a. The drawn line would be the perfect correlation (y=x). We see that there is a cluster of data points (yellow and orange and black) that lies far under that line. This means that these experimental (reduced) line widths are significantly broader than the calculated ones. When plotting these on Figure 5A, we find most of them *also* in the "lower left side" of BPTI.

This is significant: the broadened [1]HN resonances belonging to the red, orange and yellow data points in Figure 4B are

clustered and are not all over the protein. Just by itself, this result suggests that our calculation and its interpretation give rise to a useable results. But there is more. In early work, (Szyperski et al., 1993) detected [15]N exchange broadening for residues 14-16 and 38-39 in BPTI. Our current calculations (Figures 4b and 5) point to exactly the same area. (Szyperski et al., 1993) suggest that the [15]N exchange broadening is a result of the Cys14 – Cys38 disulfide isomerization at a stochastic rate of 500 s[-1] and a superposed conformational process of the entire area with a stochastic rate > 10,000 s[-1]. If

we assume that the changes in chemical shift associated with these conformational changes are the same for [15]NH and [1]HN in terms of ppm, we would expect the 500 s[-1] process to give rise to slow exchange or resonance doubling in [1]H, which is not observed. Hence it is likely that the [1]HN line widths are sensitive to the faster process.

The effect of mutations on the [15]N relaxation of BPTI has also been studied. According to (Beeser et al., 1997), fast and slow dynamics is mostly absent in wt-BPTI, with order parameters between 0.8 and 0.9 (except for the C-terminus) and

very little exchange broadening (~ 1Hz), except for two areas (again) 14-15 and 38-40. (See Figure 4A of (Beeser et al., 1997)). This is in agreement with the data of (Szyperski et al., 1993). The effect of the mutation Tyr35Gly on [15]N the relaxation parameters was also studied; it exacerbates the broadening in magnitude  (up to ~ 3 Hz) and extent (comprising residues 10-20 and 32-43), (see Figure 4B in (Beeser et al., 1997)). We show Tyr35 in Figure 5. Interestingly, our [1]H dynamic results also comprise that same *extended* area; it thus seems that the [1]HN resonances are

sensitive to extended dynamical processes present in the wild-type protein, that are only observable by [15]N relaxation after a (predictably) destabilizing mutation.

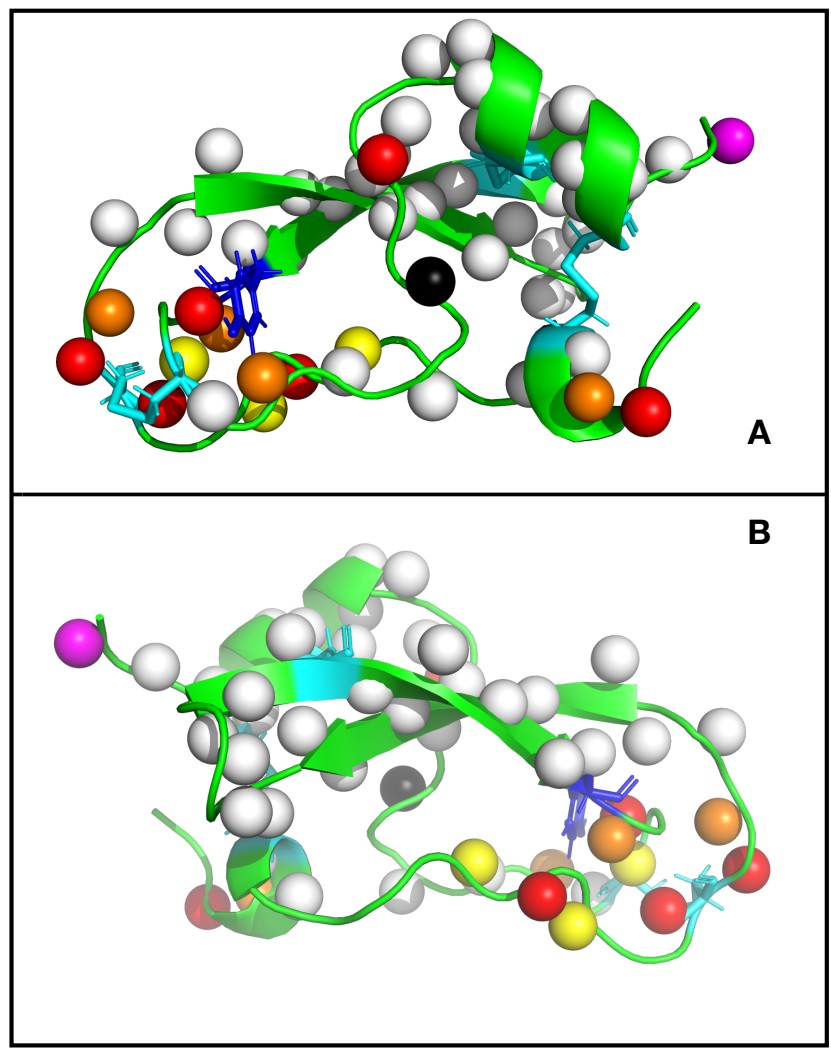

***Figure 5.*** *The crystal structure of BPTI. The spheres are the amide hydrogens. Panel B shows the molecule rotated by 180 degrees vertically. The disulfides are shown in cyan. Tyr 35 is blue. The colored amides correspond to the colored points in Figure 4 and are: Red: residues 3,11,14,15,37 and 46 ; Orange: residues 4,16,36 and 40; Yellow: residues 10,12 and 14; Black: residue 44; Magenta: residue 58 (C-terminus).*

There are no extreme outliers at the opposite side of the diagonal of Figure 4b. Such outliers would indicate resonances that according to the crystal structure coordinates should be broad (a dense proton environment), but are not broad in the experimental data. That would suggest fast local motion. In the data for Hsc70, discussed below, such outliers *are* present, and make sense in the structure. Thus, since such outliers are *not* present in the BPTI data, it suggests that the protein is rather rigid, in full agreement with the [15]N relaxation data of the wt protein (Beeser et al., 1997).



We must also address the other effects seen in Figure 5 that have no counter-part in the literature. The broadening found for two residues in the N-terminal 3-10 helix (residues 3 and 4 at the right-hand side of Figure 5A)
could be a conformational process, or amide proton mass exchange life-time broadening which is so often seen for the N-terminal 3 to 4 residues in proteins. We thus think this result is credible as well. But we have no quick rationale for the calculated broadening on Asn 44 (black) and Lys 46 (red, on the top-middle of Figure 5A). Previous work $^{15}$N relaxation work does not show anything particular for these residues. But, the resonance of K46 is actually missing at the contour level of the spectrum in Figure 1; so there is no question that something is going on there.  We may speculate that small
motions of the ring of Phe45, which hovers above amides 44 and 46, can translate in changing ring-current shifts, causing broadening for these resonances, which is not due to an actual spatial change at the level of the amides themselves. A ring-current-driven mechanism would be consistent with the lack of broadening effects in the $^{15}$N spectral data: ring current effects are, as expressed in Hz, 10 times larger for $^{1}$H  than for $^{15}$N . Hence, varying ring current shifts are apt to cause much more "conformational" exchange broadening for  $^{1}$HN than for  $^{15}$NH.
Summarizing, we find that our approach of simply calculating $^{1}$HN R$_2$ relaxation from a crystal structure can yield important dynamical data, if we concentrate on the outliers of the bulk of the distribution.

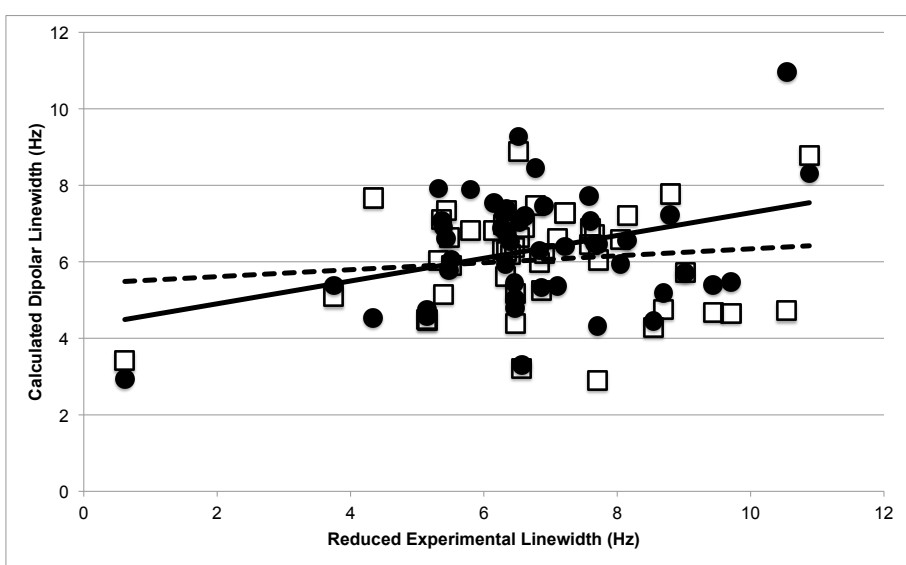

**Figure 6.** *Calculated dipolar $^{1}$HN linewidths (Eq 4 and 5) vs. the Reduced Experimental $^{1}$HN line width for BPTI. The closed circles are calculated from the crystal structure, and are also shown in Figure 4B. The drawn line is the trend line for this data with $R^2=0.122$. The open diamonds were calculated from the coordinates after minimization using the Amber ff14SB forcefield (Case et al., 2005) with implicit water. The dotted line is the trend line for this data, with a R2=0.014*

But how do we do with our calculations *within* the bulk of the distribution? Figure 6 shows that it is not good at all. There is hardly a correlation between experiment and calculation. One of the first thoughts is that the crystal structure



is, well, a crystal structure, and that the proton coordinates were added by Molprobity (Williams et al., 2018) . Maybe
some energy minimization using Amber (Case et al., 2005) would help. But it does not, as Figure 6 shows.

We concede that much improvement is needed to obtain a good correlation within the bulk of the distribution. At this point, we suggest that coordinate precision  and dynamics are the main culprits for the lack of correlation. We note that the used crystal structure with 1.2 Å resolution has a coordinate precision of ~ 0.2 Å (DePristo et al., 2004). This can give rise to considerable errors in $R_2$ calculations. For example, a HN(i) to HA(i-1) distance of nominally 2.2 Å
in a beta sheet structure (Wüthrich, 1986) may be incorrect by 9%, and produce a 68% error for the $R_2$ relaxation contribution. We wonder whether water molecules have sufficient residence times at the protein surface to contribute to dipolar relaxation of (near) surface amide protons.  At the dynamics front, there are reported variations in $^{15}$N order parameters  by 10% (Beeser et al., 1997). If the $^1$H order parameters follow this trend, this variation would be too small to cause outliers of the distribution, but be large enough to affect the correlation within the distribution. Also, at this level
of precision, relaxation interference and maybe anisotropic diffusion come back into play.  Together, we conclude that there are sufficient effects that can cause the lack of detailed correlation. Whether structure and dynamics can be untangled at this level, remains to be seen. There is work to be done here, for sure. Nevertheless, just considering the *outliers* of the distribution appears to result in relevant dynamical information.

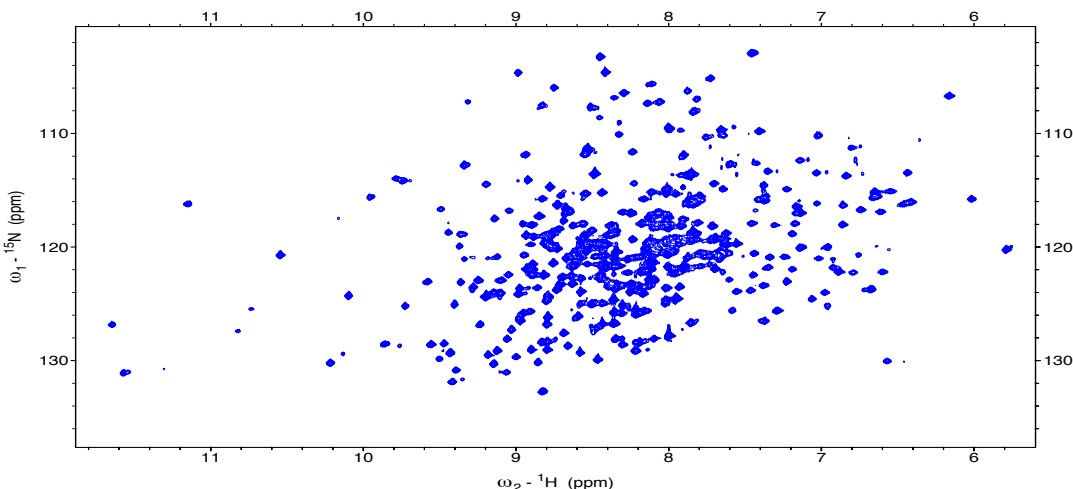

**Figure 7.** *800 MHz $^1$HN-$^{15}$N TROSY-HSQC spectrum of Hsc70 nucleotide binding domain complexed with ADP (42 kDa) processed with 1Hz EM in both dimensions.*

The impetus for this work was that we regret that detailed dynamical data can not be obtained for large proteins because the $^{15}$N relaxation experiments are not sensitive enough. The above discussion of the work with BPTI suggests that (some) of the dynamics can be learned from the $^1$HN line widths. This is encouraging. We have been interested in the family of Hsp70 chaperones for many years and in Hsc70 in particular. Hsc70 is the constitutive (i.e. present without heat shock) cellular Hsp70 which plays a major role in cellular protein homeostasis (Bukau et al., 2006). We have worked on





different domains of this protein (of different species as well), and have noticed, especially in the nucleotide-binding domain, spectral features such resonance doubling that suggest dynamical processes that may be important for the protein's function. Thus, we go ahead and do a [1]HN line width prediction from the crystal structure 3HSC.pdb, the Hsc70 nucleotide-binding domain (42 kDa) , with a resolution of 2.2 Å. A perdeuterated and uniformely [15]N, [13]C, [2]H labeled sample of this protein, recorded at 800 MHz and 303 K (see Figure 7) has been assigned (Zuiderweg and Gestwicki,

245 2017).

For a perdeuterated protein, an additional challenge for interpreting the (TROSY) HSCQ is that the amide protons may not have been completely back-exchanged in. This makes signal intensities unreliable. Second, the perdeuteration is never 100% complete. In our calculations, we approach the latter issue as follows. We scan the (crystal) structure coordinates for [1]HN, and co-add the dipolar interactions according to Eqs 3 and 4 for different spheres around the amide

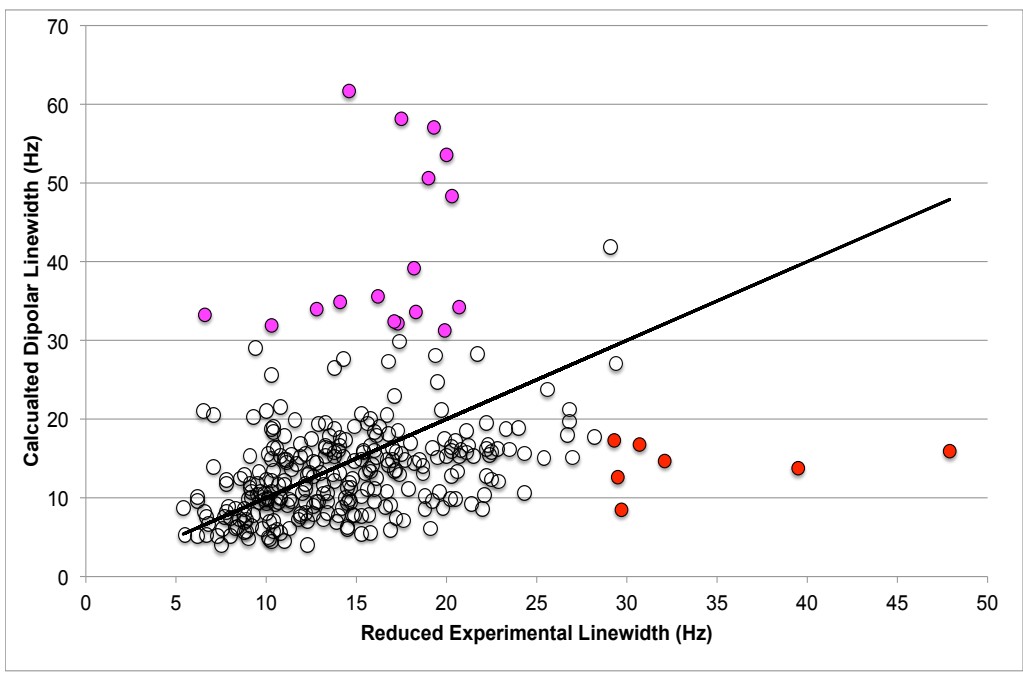

**Figure 8.** *Calculated dipolar [1]HN linewidths (Eq 4 and 5) vs. the Reduced Experimental [1]HN line width for Hsc70 NBD . Only data points for residues that are assigned and whose resonances are sufficiently resolved to be integrated in the spectrum are shown. The drawn line is y=x. The red points correspond to residues 71,177, 371, 309, 152, 175 and 340 . The magenta points correspond to residues 8, 13, 37, 44, 45, 47, 49, 58, 71, 102, 103, 105, 119, 120, 133, 211, 222, 228,241,242, 270, 296, 297, 307 and 308*

protons. We take into account all exchangeable protons (also in side chains and OH). But, we take only into account those non-exchangeable [1]HX sites in the sphere that, as determined by a random number generator, have a chance of being protonated to certain value. For each sphere, we repeat this random procedure 100 times, co-add everything and divide by 100, as if to arrive at a random deuteration pattern to a certain value for an ensemble of 100 "molecules".



Retuning back to the issue at hand, we obtain a best correspondence between experiment and calculation for 90%

perdeuteration, using a rotational correlation time of 25 ns (estimated from (Daragan and Mayo, 1997). For Figure 8 ,
we adjusted the raw experimental $^1$HN line widths by subtracting $^2$J$_{HN-Ca}$ (2 Hz), $^2$J$_{HN-CO}$ (4 Hz) (Schmidt et al., 2011)
and residual TROSY linewidth (4 Hz), as estimated from the interference of the $^1$HN CSA (Loth et al., 2005) and the $^1$H-
$^{15}$N dipolar  relaxation at 800 MHz  for $\tau_c$ =25 ns. We see that the calculated dipolar line widths span from 5 to 60 Hz.
In contrast to the data for BPTI, we find, besides  outliers under the diagonal (red), also outliers above the diagonal

(magenta). The former indicate conformational exchange broadening as in BPTI; the latter indicate a large proton density
that does *not* give rise to a broad line in the experiment – hence some fast dynamical process must have reduced that line
width. In Fig. 9 we show these outliers on the crystal structure of Hsc70 NBD. We see that most red points cluster in the
left bottom (domain IA), whereas the magenta points are more wide spread but cluster most in the top left domain
(domain IB) and less in the top right domain (IIB). The right bottom (domain IIA) is devoid of either. In the Figure 9 we

also indicate which resonances are "doubled up" or sometimes tripled up. We show only those for which we can be
reasonably sure that they constitute two conformations in slow exchange, because we *also* see a close doubling of  the
$^{13}$CO resonances in HNCO. In general, the doublets comprise 80% major and 20% minor conformations, except for the
C-terminal alpha helix at the left bottom, where the distribution is 60-40. We interpret the minor component as the ATP
conformation, which becomes predominant when adding ATP to the sample (Revington et al., 2005) as observed for a

Hsp70 of *Th. Thermophilus*. The significant issue for the current discussion is that the majority of the doubling peaks is,
like the putative exchange broadening (red dots), are also seen to be present in domain IA. So we suggest that they
represent the same conformational exchange, the latter ones with a smaller change in chemical shift. While the clustering
of exchange broadening for Hsc70 is not so evident as for the BPTI case, we are confident that it must be significant. For

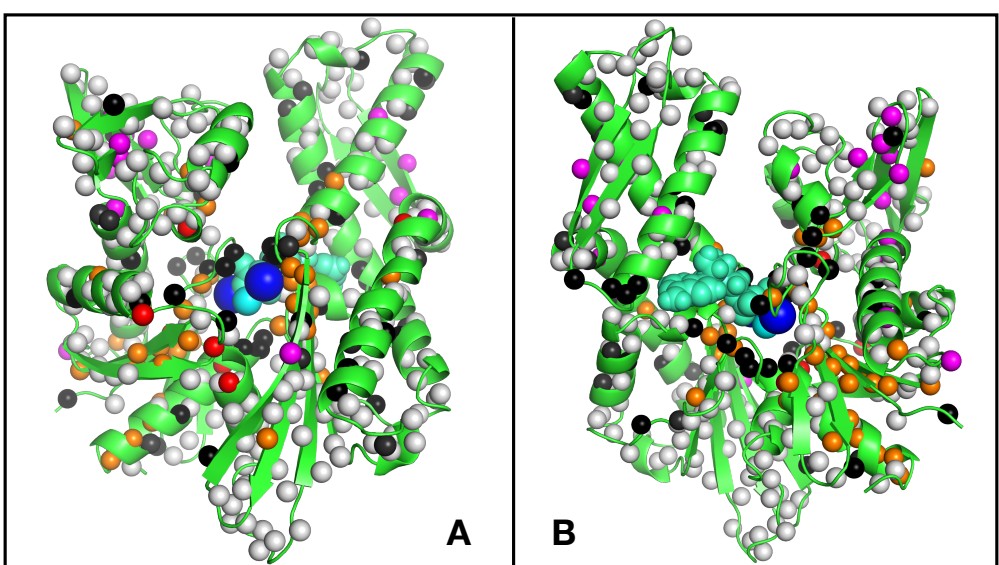

***Figure 9***. *The crystal structure of Hsc70 NBD complexed with ADP (3HSC.PDB). Panel B shows the molecule rotated by 180 degrees along the y axis as compared to panel A. The big blue spheres are Mg ions; ADP is in cyan. The smaller spheres are the amide hydrogens. The red and magenta amide hydrogens correspond to similarly colored outliers in Figure 8. Orange amides show residues that show multiple conformations in slow exchange in the spectrum. The black amides were not assigned. The grey amides have resonances that cannot be properly integrated due to overlap.*



sure, without BPTI as a test case, we would not dare to trust the Hsc70 data in this respect.

What about the magenta points, which concentrate predominantly in domain IB? We know from the RDC data on Hsc70 that the solution orientation of domain IB is tilted 10 degrees from the orientation in the crystal (Zhang and Zuiderweg, 2004). Furthermore, for a different Hsp70, DnaK of *Th. Thermophilus*, we observed that especially domain IB changes orientation between ADP and ATP (AMP-PNP) states (Bhattacharya et al., 2009). Hence, according to that data, domain IB is relatively independent of the rest of the nucleotide domain. That it would entail enhanced mobility

leading to resonance narrowing, as is suggested by the current data, is novel, but certainly not un-anticipated.

### *4. Conclusion*

We developed a computer program to predict amide proton line widths from (crystal) structures. As a calibration, we test our approach on BPTI. We find that we can predict most of the distribution of experimental amide proton line widths if

we take the dipole-dipole interaction with surrounding protons in a sphere of 6Å into account. When focusing our attention the outliers of the distribution, we find for BPTI a cluster of conformationally broadened [1]HN resonances of residues in strands 10-15 and 36-40 of the beta sheet. Conformational exchange broadening of the [15]NH resonances for these same residues was previously reported. We also apply our program to 42 kDa domain of the human Hsc70 protein. In this case, there is no previous [15]N relaxation data to compare with, but we find, again from the outliers of the

distribution, both exchange broadening and motional narrowing that appears to corroborate previous conformational insights for this domain.

### *5. Acknowledgements*

I thank Professor Dr. A. Kentgens  (Radboud University, Nijmegen, The Netherlands) for helpful discussions on solid

state NMR powder patterns.

### *6. Code/Data availability.*

The Fortran90 computer codes are available from the author.

### 300     *7. Author contributions.*

ERPZ conceived and wrote the paper. He wrote all computer codes.

### *8. Conflicting interests*

None.




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





### 10. Appendix.

### Computer program based on Eqs. 4 and 5.

The program requests a PBD file for which protons are available. Amide protons in the file should be identified as "HN",
If one wants to take into account (fractional) deuteration, then the non-exchangeables should be identified with "D" (e.g.
DB3 instead of HB3) in the PDB.

The program makes an internal copy of the pdb file. It requests the radius of the sphere of protons to consider, the
rotational correlation time, the spectrometer frequency and deuteration percentage. The code consists of three loops; the

outer loop advances over the amide protons one by one. The second loop is carried out 100 times. The innermost loop
scans the copy of the coordinates and finds all protons (including HN) around the HN at hand for the radius defined. It
decides if a particular surrounding proton is deuterated or not based on a random number process. It co-adds all $R_2$ rates
according to Eq. 4. It repeats this process 100 times, adds everything, divides by 100, and writes the results to the output
file. After that the outer loop advances to the next HN.

The program is written in Fortran90 , and contains no references to outside libraries. The source code is available from
the author.

### Computer program based on Eqs. 6 and 8 (below).

Proton-proton cross-correlated $R_2$ relaxation between just two dipolar vectors **ij** and **ik** is, adapted from (Goldman, 1984)
and (Fischer et al., 1998)


$$
R_{2CC}^{ij-ik} = \frac{1}{10} \left( \frac{\mu_0}{4\pi} \frac{\gamma_i \gamma_j \hbar}{r_{ij}^3} \right) \times \left( \frac{\mu_0}{4\pi} \frac{\gamma_i \gamma_k \hbar}{r_{ik}^3} \right)
$$

$$
\times P_2 \left( \cos\theta_{ij-ik} \right) \left\{ 9\tau_c + \frac{15\tau_c}{1+\omega_H^2 \tau_c^2} \right\}
$$

[6]

where $\theta_{ij-ik}$ is the angle between the two vectors **ij** and **ik**.

The total **$R_2$** relaxation for proton **i** is then given by


$$
R_2^{i\_total} = R_2^{ij} + R_2^{ik} \pm R_{2CC}^{ij-ik}
$$

[7]

However, these individual line widths can only be observed if the transitions for the **$H_i$** multiplet are resolved by J-
coupling (and/or residual static dipolar coupling). For amide protons, this will not be the case, and one expects an
inhomogeneous line consisting of the superposition of many narrow and broader Lorentzian lines corresponding to a

multi-spin expansion of Eq [7].

To my knowledge, there is no closed equation describing $R_2$ cross-correlated relaxation for more than two dipolar
vectors. To arrive at an estimation for the effects in a multi- proton spin system, we start from a "solid state NMR" point
of view. We calculate $B_{loc(i)}^{\Omega}$, the net local magnetic field at center proton **i** due to the surrounding protons **j** (Slichter,
1992) in certain orientation of the magnetic field with respect of the molecule:



$$B_{loc(i)}^{\Omega} = \frac{\mu_0 \hbar}{4\pi} \sum_{j \neq i}^{j=M} \Phi_{\pm} \left( \frac{\gamma_i \gamma_j}{r_{ij}^3} \right) P_2 \left( \cos \theta_{ij} \right)$$
[8]

Here, $\theta_{ij}$ is the angle between the internuclear vector $\boldsymbol{ij}$ and the magnetic field direction $\Omega$ in the molecular frame. $\Phi_{\pm}$ represents a certain configuration of the signs of the surrounding dipoles $\boldsymbol{j}$. For instance, for 10 protons one has 1024 different configurations. If one varies the magnetic field direction according to a sphere distributions and adds the results one obtains the cross-correlated powder pattern for that value of $\Phi_{\pm}$. Subsequently one co-adds all powder

patterns for different values of $\Phi_{\pm}$, and normalizes, to arrive at the "cross correlated" dipolar powder pattern for the [1]HN under consideration.

It is the time-dependence of $\boldsymbol{B_{loc}}$ as caused by molecular motion that drives the solution NMR dipolar relaxation. The $R_2$ relaxation is then obtained as the second moment of the (cross-correlated) powder pattern (Slichter, 1992):

$$R_2^{solution} = 4\tau_C \sum_{\Omega} \left( \left\langle B_{loc} \right\rangle - B_{loc}^{\Omega} \right)^2$$
[9]

where the brackets indicate average.

The computer program requires as input a "protonated" PDB file (HN for amides), the radius of the sphere of protons around the amide protons, the rotational correlation time and the spectrometer frequency. Basically, the program

consists of four nested loops: amides, protons around amides, permutation of dipole signs of these surrounding protons, and rotation of the magnetic field vector in the molecular frame.

A set of 10 nested loops permutes the dipolar signs of the closest 10 hydrogens (1024 distributions). The more remote hydrogens in the sphere (if any) have their dipolar signs assigned according to a 50% random chance.

At the inner most level, a closed loop generates an isotropic spherical distribution (5000 orientations) for the (unit)

"magnetic field" vector (http://corysimon.github.io/articles/uniformdistn-on-sphere/).

The angle between the (unit) magnetic field vector and the dipolar vector between HN and the surrounding proton is computed as the arccosine of the (normalized) dot product of those vectors.

The local dipolar field of an individual surrounding proton at [1]HN is then calculated according to Eq. 8.

This is repeated for all surrounding protons in the shell and co-added.

At this stage the program has the local field for a certain [1]HN, in a certain orientation, for a certain permutation of surrounding dipole signs. This repeated for all 5000 orientations, so that it obtains the [1]HN powder pattern for a certain permutation of the surrounding dipole signs.

Subsequently the corresponding solution NMR line width is computed from this distribution by the method of second moments (Eq. 9).

The next step is to repeat this for all 1024 permutations. The line widths are all added and normalized yielding the inhomogeneous linewidth. The inverse line widths, which are proportional to the peak height, are also added.

After that the outer loop advances to the next HN.

The program is written in Fortran90 , and contains no references to outside libraries. The source code is available from the author.





***Comparison of cross-correlated and non-cross correlated $R_2$ relaxation.***

Figure A1 shows a comparison between the line widths computed for BPTI without cross correlation (Eq [5]) and with cross correlation using the "solid state" approach (Eqs [8] and [9]) taking into account all protons within a sphere of 6 Å. While the cross-correlated results are systematically 1 Hz narrower than the ones without, one observes a strong correspondence between the "solution" and "solid state" approach. Hence, it appears justified to not include cross-

correlated relaxation in our current discussion. The effect is too small to account for the observed differences between calculation and experiment.  Perhaps taking cross correlations into account could be the final refinement when the larger perturbations arising from structural imprecision and dynamics have been untangled.

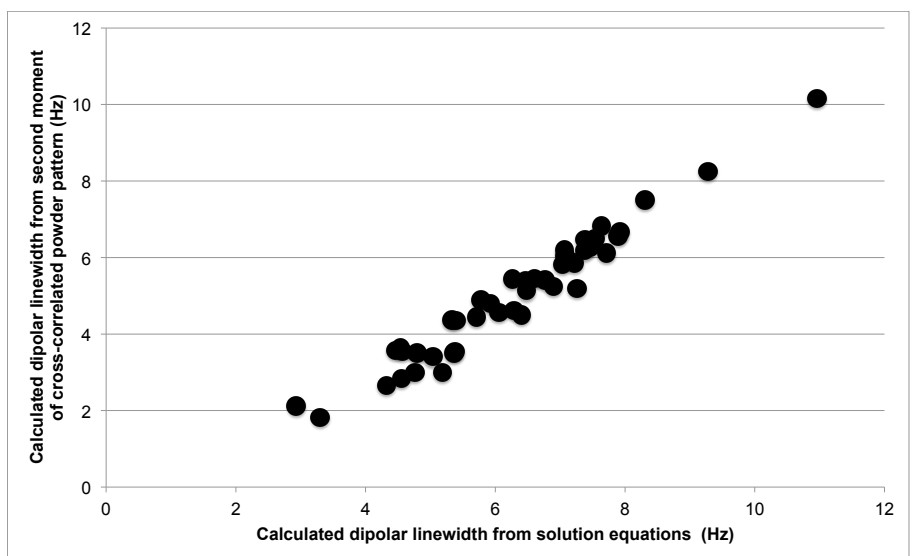

***Figure A1.***
*The effect of $R_2$ dipolar cross correlation on the $^1HN$ line widths for BPTI (9PTI.PDB). The homogeneous line widths for a shell of 6 Å as calculated from Eqs. 4 and 5 ("solution") are on the x-axis. The inhomogeneous cross correlated $^1HN$  line widths as calculated from Eqs. 8 and 9 ("powder") for the same shell are on the y-axis.*
