# Peer review of "Protein dynamics insights from 15N-1H (TROSY) HSQC"

_Magnetic Resonance, 2020_

## Author Comment (AC2)

*Dear Colleagues/Editor, here are the comments of an anonymous referee in red. My responses are in italics. As you will see, I disagree with reviewer's critiques at virtually every point.*

Reviewer: The article tries, and fails, to correlate measured line widths with line widths calculated from crystal structures, assuming dipolar relaxation as the dominant relaxation mechanism. The ultimate conclusion is that amides with narrower line widths than average are mobile, whereas amides with broader line widths reflect us – ms dynamics. This is not at all new and the conclusion in line 215 is trivial.

*This is not trivial at all. One cannot just look at the spectrum and come to that conclusion (except for very extreme outliers). The real question I am trying to address is: is a narrow line narrow because of a dilute dipolar environment, or because of fast motion?  Is a broad line broad because of exchange broadening, or because of a very dense dipolar environment?  Without taking the structure into account one cannot answer these questions.*

*Furthermore, quite a bit of hurdles must be taken: one needs to exclude amide hydrogen mass exchange, wonder about anisotropic diffusion and relaxation cross correlation. Especially the latter is not trivial at all.  Without that all being considered, one can not even attempt to calculate the line widths. With the literature data I used, one needs to also correct for unresolved $^3J_{HN\text{-}HA}$ scalar coupling (a major factor in the spread of the line width distribution: the scalar coupling varies from 1 – 10 Hz; for this small protein, it spans the same range as the bulk of the reduced experimental line width; see  Figure 4).  Only if all of this is taken into account, one can attempt to make a calculation based on the coordinates.*

*Obviously, I had also hoped that I would find a nice correlation between experiment and calculation for such a sturdy protein as BPTI, which also has a high-resolution crystal structure. But I am not getting that; it is what it is. Even if we disagree on the equations used (but see below), other equations cannot generate a correlation either – and what is suggested by reviewer would make it even worse. Most likely, coordinate precision, dynamics and conformational  exchange are the cause of the lack of correlation. It is not at all clear if all of these factors can ever be disentangled. This is a message into itself to the expert NMR community, and possibly an encouragement for new research / theory.*

*But the real value of this manuscript lies in the fact that I show that one can extract some dynamical information from the outliers in a simple HSQC. It does provide new and useful information, which can also be used by less-than experts that do not have access to sophisticated relaxation experiments.   Take a look at Figure 4B. The yellow and black points are WITHIN the main distribution of experimental values. Without the calculation these points would not be identified as broadened.  And take a look at Figure 8. The magenta points fall WITHIN the main distribution of experimental values. Without the calculations they would not be identified as motionally narrowed.*

Using M = 3 in equation 1 is inappropriate for a sensitivity-enhanced HSQC, as magnetization is not transverse throughout all delays of the PEP scheme.

*I thought I had it right. Here is the PEP scheme.*

[Figure]

*Not paying attention to signs I get*

$$\sigma_a = 2N_y H_z \cos\left(\omega_N t_1\right) + 2N_x H_z \sin\left(\omega_N t_1\right)$$

$$\sigma_b = 2N_z H_y \cos\left(\omega_N t_1\right) + 2N_x H_y \sin\left(\omega_N t_1\right)$$

$$\sigma_c = H_x \cos\left(\omega_N t_1\right) + 2N_x H_y \sin\left(\omega_N t_1\right)$$

$$\sigma_d = H_z \cos\left(\omega_N t_1\right) + 2N_z H_y \sin\left(\omega_N t_1\right)$$

$$\sigma_e = H_z \cos\left(\omega_N t_1\right) + H_x \sin\left(\omega_N t_1\right)$$

$$\sigma_f = H_y \cos\left(\omega_N t_1\right) + H_x \sin\left(\omega_N t_1\right)$$

*Between **b** and **c** both quad terms are transverse.*
*If the protein was perdeuterated, the sine term would relax significantly slower than the cosine term.*

*Between **d** and **e** the cosine term is in **z**, the sine term is transverse*
*The cosine term relaxes more slowly.*

*By the gradient EA selection, one selects for an echo in which the cosine and sine components are of equal magnitude, thus, the smallest of both pathways. So effectively, in a PFG PEP scheme, $R_2$ $^1$H relaxation determines the sensitivity during every "INEPT" step, and we have a total of 3 "INEPT" periods for the complete experiment in terms of relaxation (including the first one not shown). Not using gradients results in quad images in $f_1$.*

Equation 4 applies to indistinguishable spins. It does not apply to amide protons with distinguishable chemical shifts (see Bothner-By et al., JACS 106, 811 (1984) or equations 79 and 89 in Abragam, Chapter VIII).

*This is a very important point. I did think about this deeply; I did include much of the following in an earlier version of the manuscript, but I was recommended to simplify matters and left it out. No problem from my side to put it back in as an Appendix.*

*The $R_2$ relaxation rate for "unlike" spins such as $^1H$-$^{15}N$ is given by (Goldman, 1988)*

$$R_2^{N(H)} = \frac{1}{20}\left(\frac{\mu_0}{4\pi}\frac{\gamma_N\gamma_H\hbar}{r_{NH}^3}\right)^2$$
$$\times\left\{4\tau_c + \frac{\tau_c}{1+(\omega_H-\omega_N)^2\tau_c^2} + \frac{3\tau_c}{1+\omega_N^2\tau_c^2} + \frac{6\tau_c}{1+\omega_H^2\tau_c^2} + \frac{6\tau_c}{1+(\omega_H+\omega_N)^2\tau_c^2}\right\}$$

(1)

*where $\mu_0$ is the permittivity of space, $\gamma$ are the gyromagnetic ratios, $\hbar$ Planck's constant divided by $2\pi$, $\omega$ the resonance frequencies, and $\tau_c$ the rotational correlation time.*
*However, the $R_2$ relaxation for "like" spins, such as $^1H$-$^1H$ is more complicated:*
*One may start by replacing the $\omega_N$ terms with $\omega_H$ in Eq (1) and obtain*

$$\rho_2^{H(H)} = \frac{1}{20}\left(\frac{\mu_0}{4\pi}\frac{\gamma_H\gamma_H\hbar}{r_{HH}^3}\right)^2 \times\left\{5\tau_c + \frac{9\tau_c}{1+\omega_H^2\tau_c^2} + \frac{6\tau_c}{1+(2\omega_H)^2\tau_c^2}\right\}$$

(2)

*However, when the chemical shifts of the two $^1H$ resonances are very close, one cannot neglect a "ROESY" cross-relaxation leakage effect,*

$$\sigma_{ROE} = \frac{1}{20}\left(\frac{\mu_0}{4\pi}\frac{\gamma_H\gamma_H\hbar}{r_{HH}^3}\right)^2 \times\left\{4\tau_c + \frac{6\tau_c}{1+\omega_H^2\tau_c^2}\right\}$$

(3)

*Thus, in the literature, one adds the two terms together, just like for the Solomon $R_1$ NOE equations, and one arrives at the classical "like spins" $R_2$ equation (Goldman, 1988):*

$$R_2^{H(H)} = \rho_2^{H(H)} + \sigma_{ROE} = \frac{1}{20}\left(\frac{\mu_0}{4\pi}\frac{\gamma_H\gamma_H\hbar}{r_{HH}^3}\right)^2 \times\left\{9\tau_c + \frac{15\tau_c}{1+\omega_H^2\tau_c^2} + \frac{6\tau_c}{1+(2\omega_H)^2\tau_c^2}\right\}$$

(4)

*From a solution-NMR approach, one asks the question how close the $^1H$ frequencies have to be in order to count in the ROE effect? If they are too far apart, the coherences will become out of phase within the $R_2$ timespan, and the ROE effect will be "decoupled" away, leaving an equation (2) as correct. Indeed, by applying spin-lock the coherences will be synchronized and enable the ROE (Camelspin) to be visualized.*

*But, I am not sure if this solution intuition is the right approach. Originally, the $R_2$ relaxation rates were defined as proportional to the second-moment of the dipolar / CSA powder pattern. In solid-state NMR, the "ROE effect", or the flip-flop term is expressed as the second term of the dipolar Hamiltonian (Slichter, 1992):*

$$H_{DD}^{like} = \left( \frac{\mu_0}{4\pi} \frac{\gamma_i \gamma_j \hbar}{r_{ij}^3} \right) \times \left( I_{iz} I_{jz} + \frac{1}{4}\left( I_{i+} I_{j-} + I_{i-} I_{j+} \right) \right) \qquad (5)$$

It cannot be neglected, because (Slichter says) the dipolar interaction is much larger than the $^1H$ chemical shift differences (certainly true when that was written).
But is the dipolar coupling really that much larger? I use the leading term of Eq (5) as a measure of the "power term".

$$Dip_{1H-1H}^{like} = \frac{1}{4}\left( \frac{\mu_0}{8\pi^2} \frac{\gamma_H \gamma_H \hbar}{r_{ij}^3} \right) \quad (Hz) \qquad (6)$$

From this equation I calculate that the dipolar coupling for the canonical $^1HN$-$^1HA$ proton-pair at 2.2 A distance in a beta sheet is (+/-) 22 KHz.
The chemical shift difference between $^1HN$ and $^1HA$ is ~2000 Hz on a 500 MHz instrument. So for solid state NMR, the dipolar interaction greatly supersedes the shift difference, and therefore these are "like" spins. But for an $^1H$ and a $^1HA$ at 6A distance the dipolar coupling is just 1100 Hz . These are then more-or-less "unlike" spins for which that second term in the Hamiltonian can be neglected.
However, spins at > 6A doe not contribute much to relaxation (see Table 1 in the manuscript). So I tend to think that the "like spins" equation (4) is appropriate for those dipolar interactions due to close interactions which dominate the linewidth.
Besides from theory, I also think the proof is in the pudding. When using the "unlike" equation (2) the calculated relaxation rates would all be 5/9 of what I calculated before, and all points in Figure 4B would lie below the main diagonal. Yes, one may argue that maybe the rotational correlation time for BPTI is 6.12 ns instead of 3.4 ns, which would bring the median of the calculated distribution back to the median of experimental distribution. But this is inconsistent with at least three independent literature reports on the the rotational correlation time of BPTI using different techniques (quoted in the manuscript). Similarly, the vast majority of points in Figure 8 would lie below the main diagonal. That cannot be right either.

So, may be reviewer may be shocked that I disagree with esteemed Prof. Axel Bothner-By, not in that one would need a spinlock field to visualize the rotating frame NOE, but by the argumentation and calculations referred to above. And, as far as I can see, Abragam actually never defines what limits there are what one may call "like" spins.

As admitted by the author, Figure 4 shows no correlation. Figure 6 doesn't show a significant correlation either, again as admitted by the author. This indicates failure of the entire concept. If coordinate precision is a major problem (line 222), the concept cannot be salvaged.

No, the value of this manuscript lies in thinking about what the outliers tell us.

Line 167: The line width of the amide proton of Asp3 could very well be due to enhanced H-exchange, as the N-terminus is positively charged and the exchange base-catalyzed.

I suggested that too, on lines 205-206.

Line 187: Much more is known about the dynamics in BPTI than suggested by the manuscript, see, e.g., Grey et al., JACS 125, 14324 (2003).

It is not my purpose to review the BPTI literature.

Line 225: water molecules on protein surfaces are not known to contribute to dipolar relaxation.

*Known? What if some of them reside for a few ns? Or if many stay within a few Angstroms and form a (diverging??) dipolar field? Literature, coming from different experimental techniques, is completely at odds about these points. I know – Dave Case and I tried to publish something about this using MD and got ourselves into a hornet's nest. So, I agree, it is better to take that line out. But not because it is "known".*

Line 235: detailed dynamical data have been obtained for large proteins from methyl relaxation.

*Sure, I mentioned this at the beginning. I can mention it again. But as reviewer is undoubtedly aware, it is hard to obtain the methyl assignments (expensive labeling – many more 3D/4D experiments that are not in standard libraries) and do the methyl dynamics measurements (not in libraries either, with good reason). These difficulties are exactly the problem that keeps the dynamics insights from taking hold in the general biochemical/medical community dealing with larger proteins. My manuscript suggest a way how, without these difficulties, one can obtain at least SOME insight.*

Line 280: independent movement of a domain relative to the rest of the protein produces narrower lines. This is not novel: there are countless examples, such as calmodulin, trigger factor etc.

*The domain reorientation for one of the subdomains the 42 kDa Nucleotide-binding domain of Hsc70 was obtained using RDC experiments. Whether the reorientation is static or dynamic was not known (S/N was not good enough to obtain the magnitude of the alignment tensors with sufficient precision). The data obtained in Fig 8 shows with its magenta points that there is a dynamic component to that reorientation. This was not known. And to repeat the point I made before, the magenta points fall WITHIN the main distribution of experimental values. Without the calculations they would not be identified as motionally narrowed. And, I want to stress again, that these simple computations provide a way to obtain some dynamical information for large molecules the biochemical/biomedical world is interested in, for which relaxation experiments or RDC experiments are not practical, or lie beyond the expertise level of the research group.*

Figure 3 does not specify the acquisition times used in the spectrum - too short acquisition times would cause artificial line broadening to a variable degree for amide protons with different intrinsic line width.

*Agree – that should be mentioned. The t2 acquisition time was 136 ms. Not optimal. But it does allow a 10 Hz line (the most narrow one considered in the figure below) to decay to 1.4% of its initial value $I = I_0 \exp(-10 \times \pi \times 0.136)$ . So it is OK. I need to mention that in the past Covid-year I did not have the opportunity to collect better data myself (longer acquisition time, lower temperature so that the linewidth is more dominated by the dipolar coupling than the scalar coupling, and/or using selective pulses to eliminate the JHNHA).*

Legend of Fig. 1: EM and cosine not explained

*Agree; it should read: processed with a 1 Hz exponentional window in t2, a cosine window in t1.*

Line 57: equal to what?

*Agree, it was not clear. Equal from residue to residue.*

*These are micro seconds. Making sure they do not get misprinted.*

*Good point; this should have been explained. If I add the calculated scalar coupling to the calculated dipolar linewidth I obtain:*

[Figure]

*This is as one can see not too bad; but since it is dominated by the scalar coupling, we learn nothing with respect to dynamic line narrowing or line broadening.*

*Hsc70 exists in two conformations; one stabilized by ADP and one by ATP. The conformations are very close in free energy so that the "ATP" conformation is also present when ADP is bound. It is described in the reference given and the literature referenced therein.*

In the opinion of this reviewer, the manuscript falls far short of the criteria of Magnetic Resonance in terms of scientific impact (absence of a clear advance) or scientific quality (wrong equations used, data mostly not provided and, hence, the conclusions of the manuscript cannot be reproduced independently). It would be unexpected if it were to pass the review process of any NMR journal.

*As documented above, I disagree with reviewer in almost every respect. There is a clear advance – (some) dynamical data may become available for many more proteins. And not only the small ones most NMR spectroscopists like to work on. We disagree on the PEP scheme, but I show that reviewer is not correct. The value of the $R_2$ relaxation equations used is at discussion – I show here – part theory – part proof-in-the-pudding that my approach to the $R_2$ equations is defendable. I would love to put these arguments back into the manuscript as an Appendix. And, what does reviewer mean with "data mostly not provided " ? Everything is available on the internet. The BPTI spectrum is available at BMRB, the crystal structure at the PDB, and the program, simple as it is, after I figured out that it does not have to be more complicated than that, can be provided by me. I guess very few would be interested to take up Hsc70.*

---

## Author Comment (AC3)

*Dear Colleagues/Editor, here are the comments of an anonymous referee #2 in red. My responses to the reasonable comments are in italics.*

This is a largely exploratory manuscript investigating the potential to obtain (semi-)quantitative results on protein dynamics based on backbone amide 1H linewidths in 1H-15N HSQC or TROSY spectra. As Zuiderweg states himself, the underlying idea to glean dynamic information from the HSQC spectrum is commonly used by NMR spectroscopists in a qualitative sense by identifying particularly sharp or broadened cross-peaks. Here, Zuiderweg provides a more comprehensive analysis by attempting to account for all contributions to the 1HN linewidth from various relaxation mechanisms and other effects, such as unresolved J-couplings and B0 inhomogeneity. Naturally, the dominant part of the relaxation is due to dipole-dipole relaxation, which is estimated for each backbone amide based on the high-resolution crystal structure of the protein in question. The end result is that the calculated linewidths show only fair agreement with the experimental ones, but outliers appear to reliably identify backbone amides that are either undergoing large-amplitude fast internal dynamics (i.e., residues with low order parameters) or residues undergoing conformational exchange. Thus, we are left with the conclusion that 1HN linewidths cannot provide more detailed information than what is customarily obtained from a qualitative, first-glance interpretation of HSQC-type spectra. To this extent, the work clearly does not advance the field since it does not provide any substantial conclusions beyond current knowledge. Still, I appreciate the comprehensive, semi-quantitative analysis offered by Zuiderweg, which clearly shows the limitations of the proposed analysis. In essence, the work demonstrates that 1HN linewidths cannot be interpreted in terms of dynamics to any detailed extent. For these reasons, I believe that the study could be worth publishing.

*I do agree with reviewer the work demonstrates that 1HN linewidths cannot be interpreted in terms of dynamics to any detailed extent. This is indeed one important outcome of the work: a "heads up" that there is still challenging (theoretical/computational/experimental) work to be done in solution structural biology.*

*But there is another part; the real question I am trying to address is: is a narrow line narrow because of a dilute dipolar environment, or because of fast motion? Is a broad line broad because of exchange broadening, or because of a very dense dipolar environment? Without taking the structure into account one cannot answer these questions. Thus it takes more than just looking at the spectra: and, one needs to correct for unresolved 3JHN-HA scalar coupling (a major factor in the spread of the line width distribution: the scalar coupling varies from 1 – 10 Hz; for this small protein, it spans the same range as the bulk of the reduced experimental line width; see Figure 4). Now one can compare the reduced linewidth with calculations. Why calculations? Take a look at Figure 4B. The yellow and black points are WITHIN the main distribution of experimental values. Without the calculation these points would not be identified as broadened. And take a look at Figure 8. The magenta points fall WITHIN the main distribution of experimental values. Without the calculations they would not be identified as motionally narrowed. Thus the work advances the field beyond current knowledge.*

Minor points:

p. 4, Eq [1]: I do not follow this equation fully: the last two factors are not defined and it is not clear why they appear in the equation.

*Indeed, there is an error. The equation should read*

$$S/N \sim \left\{ \exp\left( -\pi \delta v_{1/2}^{HN} / \left( 2 \times {}^{1}J_{HN} \right) \right) \right\}^{M} \times \exp\left( -\langle t_1 \rangle \delta v_{1/2}^{N} \right) \times \frac{1}{\delta v_{1/2}^{N}} \times \frac{1}{\delta v_{1/2}^{HN}}$$

*S/N is defined as peak height / noise. Peakheight is also proportional to the inverse linewidth.*

The first exponential assumes that each step in the reverse polarization transfer (after t1) contributes equally to the linewidth, but this is not true for all pulse sequences (it depends on the details of the PEP scheme, etc).

*I thought I had it right. Here is the PEP scheme.*

[Figure]

*Not paying attention to signs I get*

$$\sigma_a = 2N_y H_z \cos(\omega_N t_1) + 2N_x H_z \sin(\omega_N t_1)$$

$$\sigma_b = 2N_z H_y \cos(\omega_N t_1) + 2N_x H_y \sin(\omega_N t_1)$$

$$\sigma_c = H_x \cos(\omega_N t_1) + 2N_x H_y \sin(\omega_N t_1)$$

$$\sigma_d = H_z \cos(\omega_N t_1) + 2N_z H_y \sin(\omega_N t_1)$$

$$\sigma_e = H_z \cos(\omega_N t_1) + H_x \sin(\omega_N t_1)$$

$$\sigma_f = H_y \cos(\omega_N t_1) + H_x \sin(\omega_N t_1)$$

*Between **b** and **c** both quad terms are transverse.*
*If the protein was perdeuterated, the sine term would relax significantly slower than the cosine term.*

*Between **d** and **e** the cosine term is in z, the sine term is transverse*
*The cosine term relaxes more slowly.*

*By the gradient EA selection, one selects for an echo in which the cosine and sine components are of equal magnitude, thus, the smallest of both pathways. So effectively, in a PFG PEP scheme, $R_2$ $^1H$ relaxation determines the sensitivity during every "INEPT" step, and we have a total of 3 "INEPT" periods for the complete experiment in terms of relaxation (including the first one not shown). Not using gradients results in quad images in $f_1$.*

l. 94: aromatic ring flipping does not cause exchange linebroadening of amide protons (since the end states are identical).

*Reviewer is correct.  The ring protons themselves can be broadened, but not anything else.*

Table 1: Please clarify what is listed in this table. By comparing with Fig. 4, I assume that the Sum of 1HN linewidths is taken over all residue pairs in the protein(?). This should be stated in the Table header (or footnote).

*Indeed this could be clearer. I will also change it to average calculated 1HN linewidth.*

Figure 6 legend: "open diamonds" should be 'open squares'.

*Thanks*

The text should be checked for typos, incorrect order of words, missing words, etc.

---

## Author Comment (AC4)

*Dear Colleagues/Editor, here are the comments of an anonymous referee #3 in red. My responses to the reasonable comments are in italics.*

The manuscript proposed by Erick Zuiderweg presents a valuable attempt to rationalize the intensities measured on protein 1H-15N correlation spectra in order to get a qualitative description of the backbone dynamics at different timescales. The different parameters affecting the 2D correlation peak's intensities are exhaustively listed and their values estimated from a structural model of the protein. The concept is tested on the BPTI protein and, as noted by the author, modelling the different known contributions to the signal intensities fails to reproduce the experimental measurements. The relevance of the approach is defended by the observation that large deviations from the modelled intensities do cluster in regions of BPTI where specific dynamical features where previously reported from 15N relaxation measurements. A graphic based clustering of these "model deviating" signals is therefore proposed as a fast approach to get qualitative insights on the protein dynamics. The approach is applied to a large protein (Hsc70) enabling some observations to be made on its dynamical properties.

As stated by the author in his introduction, such an approach would be very valuable in the field of protein NMR, as we do share the general feeling that the information content of 15N-HSQC or TROSY are under-exploited, and the author's attempt to address this task is interesting. However, my general opinion is that the current state of this development is far too preliminary and would desserve more work to be published. The general applicability of such a method should be assessed by probing the concept on different class of proteins that display distinct and well documented dynamical features (depending on their size, geometry, experimental conditions of the study T° pH ...). The "correlation" approach is indeed the only way to go when the model fails to reproduce the experiment. Anyhow, some statistical assessment is necessary for the applicability of the proposed approach on other systems: for instance, what criteria should be used to identify a residue with abnormal intensities ? A quantitative description of the deviating between the theoretical model and the experimental values is clearly missing here.

*Indeed, the work is crying out for follow-ups. It is a "heads up" that there is still challenging (theoretical/computational/experimental) work to be done in solution structural biology. One first thing to do is to collect data for e.g. BPTI, Ubiquitin and GB3 at low temperature so that the relative contribution of the scalar coupling is reduced. I may invite colleagues to send me such data or collect it myself as soon as the pandemic is over. And yes, on the basis of better data, more quantitative characterizations of "outliers" can be formulated. Second is to explore if and how we can adapt (improve??) the structures to correspond better with the linewidths. I am already planning that with a colleague. I hope others will follow. For now, with the deadline of Festschrift, it will have to wait until this manuscript is published.*

Some of the hypothesis made by the author are questionable. In particular, the assumption that intrinsic exchange rates are identical for all amide protons is probably not true since local chemical environment at the protein surface do modulate these exchanges with water. Such information may be obtained by simple 2D experiments such as the Het-SOFAST proposed by Paul Shanda and Bernhard Brutscher.

*The unprotected amide proton mass exchange rate is calculated to be 1.15 $s^{-1}$ exchange rate, giving rise to a broadening of ~ 0.3 Hz for unprotected amide proton resonances. Protected amides will exchange (much) slower. While reviewer is in principle correct that environment matters, it would in this case only affect differences in rates that are slower than 1 s-1, thus of no relevance to the (precision of ) the current data.*

Relaxation mechanisms different from the dipole-dipole interaction may also contribute for the amide transverse relaxation: can we fully discard scalar relaxation ?

*This is an interesting point to keep in mind for later. The 3JHNHA will fluctuate when the dihedral angle phi fluctuates. Depending on the timescale, this may contribute to broadening. For now, however, we have left dynamics completely outside of the scope of this paper and we are exploring if we can calculate the linewidths on the basis of a static structure. The answer is that we cannot do that with any precision (but outliers are*

*valuable). As I stated on line 222: "At this point, we suggest that coordinate precision and dynamics are the main culprits for the lack of correlation."*

Small points:

- Equation 1 contains some mistakes:

I guess the two factors in the denominator are line-width (delta_nu) and not frequencies (delta missing)

*Yes, thanks.*

$$S/N \sim \left\{\exp\left(-\pi\delta\upsilon_{1/2}^{HN}/\left(2\times{}^{1}J_{HN}\right)\right)\right\}^{M}\times\exp\left(-\langle t_{1}\rangle\delta\upsilon_{1/2}^{N}\right)\times\frac{1}{\delta\upsilon_{1/2}^{N}}\times\frac{1}{\delta\upsilon_{1/2}^{HN}}$$

The term describing the nitrogen relaxation doesn't make sense to me: why is the average <t1> considered ? The amplitude of the peak on the F1 (15N) dimension depends on the magnetisation's level at the end of the t1 increment.

*Of course. But when t1 is short the transfer will be large, and when t1 is long, the transfer will be smaller, and some average determines the overall transfer efficiency.*

-140 Please describe how the amide value of CSA is derived ? Is it reasonable to assume the same CSA for all amides ? could this not be one major reason of the observed deviations, since the amide may be engaged within a hydrogen bond modulating the distribution of electrons.

*I have taken the amide CSA as measured for Ubiquitin in Figure 3, right column, second panel by Loth et all.*

[Figure]

*As one sees, the individual tensor principal values vary quite a bit.*

*For R2 CSA relaxation (rhombic tensor) one has*

$$\frac{1}{T_{2}^{CSA-non-axial}}=\frac{(\omega_{I}\Delta\sigma)^{2}}{18}\left\{1+\frac{\Delta\eta^{2}}{3}\right\}\times\left\{4j_{0}(0)+3j_{-1}(\omega_{I})\right\}$$

*where, with $\sigma_{11}\geq\sigma_{22}\geq\sigma_{33}$*

$$\Delta\sigma = \sigma_{11} - \frac{\sigma_{22} + \sigma_{33}}{2}$$

$$\Delta\eta = \frac{\sigma_{22} - \sigma_{33}}{\sigma_{11} - \sigma_{iso}}$$

$$\sigma_{iso} = \frac{\sigma_{11} + \sigma_{22} + \sigma_{33}}{3}$$

*I calculate from the values in Figure above a variation 0.4 Hz to 0.08 Hz in 1H CSA contribution to line width for tc=3.5 ns, on a 500 MHz spectrometer. Yes, it varies a bit, but the values are really small.*

*I will change the statement*

*"We estimate that the [15]N-[1]HN dipolar interaction accounts for 3 Hz, that the [1]H CSA contributes 1 Hz at 500 MHz, while field inhomogeneity typically is limited to 1 Hz. What the exact values are may be disputed, but they are small and approximately constant for all amides, or partially cancel in the TROSY version of the HSQC."*

*To*

*"We estimate that the [15]N-[1]HN dipolar interaction accounts for 4 Hz, and should be constant for all amides. The [1]H CSA varies a bit , but contributes less than 1 Hz to the 1HN linewidth at 500 MHz for this small molecule, while field inhomogeneity typically is limited to 1 Hz. "*

145 The author should mention the model they used to derive the linewidths from Sparky (Lorentzian fitting ? gaussian ?)

*For the BPTI HSQC I found that Lorentzian was best, while for the Hsc70 TROSY Gaussian was best. I will add this to the legends.*

- Figure 4: There are some discrepancies between the text and the figure:

   - orange points are below 11 Hz for Reduced Experimental Line width

*Would the following be clearer? " The orange points all have a reduced experimental line width of less than 11 Hz"*

   - What is meant by "at opposite side of the diagonal ?" I suggest using "Upper triangle" and lower triangle regions

*Yes, I like that suggestion.*

-      Figure 6:

         - Legend : plain circle and squares

*Agree. Should read "open squares"*

- Figure 9 : Labelling the different domains of Hsc70 would be helpful to follow the dynamical description.

*Agree. Will do.*